# An Evolutionary Perspective on the Origin, Conservation and Binding Partner Acquisition of Tankyrases

**DOI:** 10.3390/biom12111688

**Published:** 2022-11-15

**Authors:** Sven T. Sowa, Chiara Bosetti, Albert Galera-Prat, Mark S. Johnson, Lari Lehtiö

**Affiliations:** 1Faculty for Biochemistry and Molecular Medicine & Biocenter Oulu, University of Oulu, 90220 Oulu, Finland; 2Structural Bioinformatics Laboratory, Biochemistry, Faculty of Science and Engineering and InFLAMES Research Flagship Center, Åbo Akademi University, 20520 Turku, Finland

**Keywords:** tankyrase, PARP, ARTD, ADP-ribosylation, protein evolution, SLiM, LMB-hub

## Abstract

Tankyrases are poly-ADP-ribosyltransferases that regulate many crucial and diverse cellular processes in humans such as Wnt signaling, telomere homeostasis, mitotic spindle formation and glucose metabolism. While tankyrases are present in most animals, functional differences across species may exist. In this work, we confirm the widespread distribution of tankyrases throughout the branches of multicellular animal life and identify the single-celled choanoflagellates as earliest origin of tankyrases. We further show that the sequences and structural aspects of TNKSs are well-conserved even between distantly related species. We also experimentally characterized an anciently diverged tankyrase homolog from the sponge *Amphimedon queenslandica* and show that the basic functional aspects, such as poly-ADP-ribosylation activity and interaction with the canonical tankyrase binding peptide motif, are conserved. Conversely, the presence of tankyrase binding motifs in orthologs of confirmed interaction partners varies greatly between species, indicating that tankyrases may have different sets of interaction partners depending on the animal lineage. Overall, our analysis suggests a remarkable degree of conservation for tankyrases, and that their regulatory functions in cells have likely changed considerably throughout evolution.

## 1. Introduction

Tankyrases (TNKSs) are part of the ARTD enzyme family and they catalyze a sequential transfer of ADP-ribose from NAD^+^ to their protein substrates leaving them modified with poly-ADP-ribosyl (PAR) chains [1,2]. TNKSs regulate a wide variety of pathways and physiological processes, such as telomere maintenance [3,4], glucose metabolism [5,6], bone growth [7], mitotic spindle formation [8,9], Wnt/β-catenin signaling [10] and Hippo/YAP-signaling [11]. Humans and other vertebrates possess two highly similar TNKS paralogs: TNKS1 and TNKS2. It is yet unclear if these paralogs have different functions in the cell; however, they are at least redundant to some degree as indicated by knockout experiments in mice [12]. Among ARTD members, which are also referred to as PARPs [13], TNKSs have a unique domain architecture (Figure 1a). In addition to their catalytic ADP-ribosyltransferase (ART) domain, TNKSs have five N-terminal ankyrin-repeat (ANK) domains, which are also termed ankyrin repeat clusters (ARCs) and they are numbered ARC1 to ARC5. With the exception of ARC3, which may have a structural role [14,15,16], the ARCs bind interaction partners of TNKSs. These interaction partners bind via a small linear TNKS binding motif (TBM). TBMs have a canonical sequence Rxx[ACGP]xGxx, although also less constrained variations such as Rx(4)Gxx and Rx(5)Gxx were reported to bind to TNKSs [10,15,16,17]. A sterile alpha motif (SAM) domain is located between the ARC and ART domains and mediates the multimerization of TNKSs [18,19,20,21]. In contrast to TNKS2, TNKS1 has an additional and likely disordered N-terminal extension of unknown function which is rich in histidine, proline and serine residues and was accordingly termed an HPS region [22].

In the current model, PARylation of proteins binding to TNKSs leads to their ubiquitination and subsequent proteasomal degradation (Figure 1b). Additionally, TNKSs may act as scaffolding proteins by providing a platform for protein–protein interactions [19,21,26]. Importantly, the regulatory functions that TNKSs exert on different pathways in the cell are a consequence of specific binding partners interacting with TNKSs. Numerous proteins interacting with TNKSs have been reported in humans, although it has become clear that these interactions are conserved in varying degrees between species (Figure 1c). For example, TNKSs regulate telomere length homeostasis through the interaction with the telomeric protein TRF1 [4,27]. This interaction and thus the function of TNKSs in telomere homeostasis is absent in mice due to a missing TBM in the mouse TRF1 ortholog [23,28]. In contrast, the function of TNKSs as positive regulators of Wnt signaling through interaction and regulation of Axin levels appears well-conserved across species, and presence of the TBM sequence in Axin orthologs was reported even in invertebrates [10]. The regulatory function of TNKSs on Wnt signaling was shown in several species such as human [10], mouse [29], zebrafish [30] and fruit fly [31].

While many interaction partners of TNKSs have been confirmed in humans [1,32,33,34], the presence of TBMs in orthologs of other species was only shown in some cases and to varying degrees in terms of taxonomic coverage (Figure 1c). In this work, we examined the distribution of TNKSs across species, the conservation in terms of sequence and structure as well as the conservation of the presence of TBMs in orthologs of 21 human binding partners. We show that TNKSs are widely distributed throughout multicellular animals and we further identify single-celled choanoflagellates as the likely earliest origin of TNKSs. We demonstrate further that TNKSs display a high degree of conservation and show through in vitro experiments with an early diverged TNKS ortholog from the demosponge *Amphimedon queenslandica* that the basic molecular functions of TNKSs are also highly conserved. On the other hand, the interaction with different TNKS binding partners appears to be far less conserved, as the presence of TBMs in orthologs varies greatly among different metazoans.

## 2. Materials and Methods

### 2.1. Sequence Acquisition

Canonical sequences from human orthologs were obtained from UniProt. Sequences from non-human orthologs were obtained from NCBI non-redundant (NR) database using sequences of human orthologs as a query by BLAST (http://blast.ncbi.nlm.nih.gov/Blast.cgi (accessed on 31 October 2022)) [35]. If no ortholog could be identified in protein databases, the tBLASTn implementation against the Genebank database (TSA and WGS) was used. Sequences of TNKSs from *E. burgeri* and *M. leidyi* were acquired from the Ensembl database (https://metazoa.ensembl.org/ (accessed on 20 April 2022)). Possible functional annotations for the TNKS-like protein from *N. fowleri* were examined in the Amoeba database (https://amoebadb.org/ (accessed on 20 April 2022)) under the gene identifier NF0130940.

### 2.2. Sequence Alignments and Construction of Phylogenetic Trees

Multiple sequence alignments were conducted with Clustal Omega [36] or the Clustal W implementation in the MEGA11 software [37]. Global pairwise sequence alignments were conducted with EMBOSS Needle and local pairwise alignments were conducted with EMBOSS Matcher. The construction of phylogenetic trees was conducted with the maximum likelihood and neighbor joining methods in MEGA11 [37] using the Jones–Taylor–Thornton (JTT) substitution model [38] and a bootstrap analysis with 1000 replicates was performed for each tree.

### 2.3. Structure Visualization and Prediction

Models of experimentally derived protein structures were obtained from the Protein Data Bank (PDB) [39]. The structure models were visualized in the PyMOL Molecular Graphics System (PyMOL, version 1.8.4.0). The prediction of protein structures from *A. manzaensis* and *N. fowleri* were conducted with RoseTTAFold [40] using only the protein sequences as input without additional constraints.

### 2.4. Position-Specific Sequence Conservation Analysis of Tankyrases

Sequences of TNKSs originating from a broad taxonomic range were sampled and obtained as described above. Sequences annotated as low-quality proteins were removed and the remaining sequences were inspected for obvious indications of assembly errors. A multiple sequence alignment from 113 TNKS sequences was used as input for residue specific conservation analysis by ConSurf [41]. JTT was determined to be the best evolutionary model for the data provided and was used by ConSurf to estimate the evolutionary relationships. ConSurf conservation scores were calculated based on the Bayesian method.

### 2.5. Identification of Potential TBMs in Orthologs of TNKS Binders

A list of 21 human TNKS binding partners was manually curated based on reports in the literature. Sequences for the orthologs of the human TNKS interaction partners were obtained as described above. A multiple sequence alignment was performed for each set of orthologs, and the presence of potential TBMs in respect to the human ortholog was examined for the proteins in each set. The different configurations Rxx[ACGP]xGxx, Rx(4)Gxx and Rx(5)Gxx were accepted as possible TBMs. Generally, only those TBMs that aligned with the position of the TBM in the human ortholog were considered potential TBMs; although not aligned, some possible TBMs were annotated due to presence in a likely disordered region in proximity to the TBM in human orthologs. For RNF146, all possible TBMs were located in the protein sequences of the orthologs. The positions of domains in the RNF146 orthologs were predicted using InterPro [42].

### 2.6. Molecular Cloning

The cloning of human TNKS1 constructs was conducted as previously described for CFP-ARC5 [43] or SAM-ART and ART domains [44]. Cloning of the YFP-REAGDGEE construct was conducted as previously described [43]. The *E. coli* codon-optimized constructs for *A. queenslandica* TNKS (*Aq*TNKS: XP_019848937.1) were obtained as gBlock from IDT. The construct ARC5 (*Aq*TNKS Glu658-Met811) was cloned into the pNIC-CFP expression vector (Addgene #173074). The *Aq*TNKS constructs SAM-ART (Pro889-Thr1181) and ART (Thr968-Thr1181) were cloned into the pNIC-MBP vector, which was prepared as previously described [43]. Cloning was conducted using the SLIC method [45]. Briefly, the vectors were linearized and mixed with insert and T4 DNA polymerase. The reaction mixture was used to transform chemicompetent *E. coli* NEB 5α cells (New England BioLabs). Colonies were grown at 37 °C on LB agar plates containing 5% sucrose and 50 μg/mL kanamycin; genes in the vector encoding kanamycin-resistance and SacB enzyme [46,47] served as selection markers for successful transformation and vector linearization, respectively. All constructs were verified by sequencing of the insert regions.

### 2.7. Protein Expression

Chemicompetent *E. coli* BL21(DE3) cells were transformed with the expression vectors. For each construct, 500 mL Terrific Broth (TB) autoinduction media including trace elements (Formedium, Hunstanton, Norfolk, England) were supplemented with 8 g/l glycerol and 50 μg/mL kanamycin and inoculated with overnight preculture. The flasks were incubated at 37 °C (shaking) until an OD_600_ of 1 was reached. The incubation continued overnight at 18 °C (shaking). At the end of the incubation period, the cells were harvested by centrifugation at 5000× *g*, 4 °C for 30 min. The pellets were resuspended in lysis buffer (50 mM HEPES pH 7.5, 500 mM NaCl, 0.5 mM TCEP, 10 mM imidazole, 10% glycerol *v*/*v*) and used immediately for purification or frozen at −20 °C.

### 2.8. Protein Purification

Purification of human TNKS1 ART and TNKS1 SAM-ART [44] or CFP-ARC5 (TNKS1) and YFP-REAGDGEE [43] was conducted as previously described. Frozen cells were thawed first. The cells were supplemented with 0.1 mM Pefabloc SC (Roche) and 20 μg/mL DNase I (Roche) and lysed by sonication. The lysate was centrifuged (30,000× *g*, 4 °C, 30 min) and filtered (0.45 μm). As initial purification step, all the proteins were purified by immobilized metal affinity chromatography (IMAC) with a 5 mL HiTrap HP column equilibrated with lysis buffer and charged with Ni^2+^. Lysis buffer with higher concentrations of imidazole was used for washing steps (25 mM) and elution (300 mM).

The IMAC purification of the *Aq*TNKS ART domain (MBP-tagged) was followed by buffer exchange with size-exclusion chromatography (SEC) buffer (20 mM HEPES pH 7.5, 350 mM NaCl, 0.5 mM TCEP, 10% *v*/*v* glycerol) using an Amicon Ultra-15 centrifugal filter (30 kDa MWCO). To remove MBP, the sample was incubated with TEV protease (1:30 molar ratio) for 48 h at 4 °C and was then run over a 5 mL MBPTrap HP column equilibrated with SEC buffer. SEC was performed as last step of the purification using an S75 16/600 size-exclusion chromatography column and SEC buffer.

For *Aq*TNKS SAM-ART (MBP-tagged), the IMAC step was followed by MBP affinity chromatography using a 5 mL MBPTrap HP column equilibrated with SEC buffer. After loading, the column was washed with 5–10 column volumes of SEC buffer and eluted with SEC buffer containing 10 mM maltose.

For the *Aq*TNKS ARC5 domain (CFP-tagged), the elution from IMAC was purified by SEC as above. To remove contamination by *E. coli* chaperone proteins, a successive IMAC purification was performed and 5 mM MgCl_2_ and 5 mM ATP disodium salt were included in the washing step [48]. Finally, a second purification by SEC was performed as above. At the end of each purification, all the proteins underwent concentration, they were aliquoted, flash frozen in liquid nitrogen and stored at −70 °C.

### 2.9. Activity Analysis by Western Blot

The constructs of TNKS1 or *Aq*TNKS were prepared in absence or presence of a mixture of biotinylated-NAD^+^/NAD^+^ (1:10). Additionally, a control with PARG (0.2 mg/mL) was prepared. Final total NAD^+^-concentrations were 2 μM and 10 μM for TNKS1 and *Aq*TNKS, respectively. The reactions were incubated for 1 h at room temperature. At the end of the incubation period, 10 μL of each reaction were run on SDS-PAGE (Mini-Protean TGX 4–20% gradient gel, Bio-Rad). The transfer of the proteins to a nitrocellulose membrane (Trans-Blot Turbo Mini 0.2 μm Nitrocellulose Transfer Pack, Bio-Rad) was performed using a Trans-Blot semi-dry system (Bio-Rad). Afterwards, the membrane was washed with TBS-T and subsequently stained with Ponceau S solution. The membranes were imaged (ChemiDoc Imaging System, Bio-Rad) and the staining was removed by washing the membrane in TBS-T. The membranes were blocked using 1× TBS with 1% casein (Bio-Rad) for 30 min. The membranes were incubated with streptavidin-HRP (PerkinElmer) diluted 1:10,000 in blocking buffer for 30 min. The membranes were washed with TBS-T buffer and HRP detection was conducted using ECL solution (Advansta). Following the imaging step, the membranes were washed in TBS-T containing 5% (*w*/*v*) skimmed milk powder for 15 min. The membranes were transferred to blocking solution including 0.1 μg/mL nanoluciferase-ALC1 [49] for 30 min. Finally, the membrane was washed in TBS-T and detection of nanoluciferase on the membrane was conducted with 1:500 NanoGlo (Promega) diluted in PBS buffer.

### 2.10. Analytical Size-Exclusion Chromatography

In separate independent runs, 0.5 mg of each protein sample (*Aq*TNKS SAM-ART, *Aq*TNKS ART, TNKS1 SAM-CAT and MBP) was loaded to a Superdex 200 Increase 10/300 GL column. The column was equilibrated in SEC buffer (20 mM HEPES pH 7.5, 350 mM NaCl, 0.5 mM TCEP, 10% glycerol) and samples were run at a flow rate of 0.5 mL/min at 4 °C. The absorbance at 280 nm was recorded. In each run, 1 mL fractions were collected. Fractions corresponding to the different peaks observed for the SAM-ART constructs were analyzed by SDS-PAGE.

### 2.11. Differential Scanning Fluorimetry

Samples of ART constructs (5 μM) from TNKS1 or *Aq*TNKS were prepared in absence or presence of TNKS inhibitors (25 μM) IWR-1, G007-LK, OM-153, compound 40, XAV939, E7449 or NVP-TNKS656 [50] and 5× SYPRO orange dye (ThermoFisher Scientific). Each condition was prepared in 4 replicates with a final volume of 20 μL. The sample buffer was 10 mM Bis-Tris-Propane (pH 7.0), 3% PEG 20,000 (*w*/*v*), 0.01% Triton X-100 (*v*/*v*), 0.5 mM TCEP. The samples were transferred into a 96-well transparent qPCR plate and the measurements were performed using a C1000 CFX96 thermal cycler (Bio-Rad). The temperature was increased from 20 °C to 95 °C (1 °C/min) and data points for melting curves were recorded in 1 min intervals. The data analysis was performed with GraphPad Prism 9 using a nonlinear regression analysis (Boltzmann sigmoid equation) of normalized data.

### 2.12. Determination of the Dissociation Constants by FRET

The experiment was performed as previously described [43]. Briefly, CFP-fused ARC5 domains (100 nM) from TNKS1 or *Aq*TNKS were mixed with YFP-REAGDGEE (0–1500 nM). The FRET emission for each condition was determined. The calculation of dissociation constants was conducted using a method described by Song et al. [51].

## 3. Results

### 3.1. Distribution and Origin of Tankyrases

Species that are frequently mentioned in this manuscript are shown in Table 1. We started our analysis by characterizing the distribution of tankyrases throughout the tree of life (Figure 2a). The presence of TNKSs was searched within major phylogenetic groups. An early study on the evolutionary history of PARP proteins by Citarelli et al. only found canonical TNKSs present in bilaterians, while Perina et al. later identified TNKSs also in cnidarians and sponges, concluding that TNKSs may be confined to metazoans [52,53]. Indeed, we found and confirmed the presence of at least one TNKS homolog in all vertebrates analyzed. While TNKS homologs are present in early diverged bilaterians such as the fruit fly *Drosophila melanogaster* and the mollusk *Octopus bimaculoides*, no TNKS homolog was found in the nematode *Caenorhabditis elegans*. The absence of TNKSs and other PARP homologs in *C. elegans* was reported previously, and the loss of these PARP genes was discussed [52]. It is worth mentioning that TNKSs appear to be present in other nematodes such as *Brugia malayi* and *Trichinella spiralis*.

Furthermore, our search confirmed that TNKSs are present in early diverging metazoan species such as the sponge *Amphimedon queenslandica*, the cnidarians *Hydra vulgaris* and *Nematostella vectensis* and the ctenophore *Mnemiopsis leidyi*. However, we could not identify a TNKS homolog in the placozoan *Trichoplax adhaerens*. While the exact taxonomic placement of placozoans is still debated, it is widely accepted that they diverged after the appearance of sponges [54,55], which indicates a loss of TNKS genes in this lineage similarly to *C. elegans*. In fact, it was suggested that many physiological traits such as muscles, nerves and guts were lost in placozoans in favor of a simpler lifestyle [54,56], which likely resulted in the loss of many genes.

It has been suggested that the gene duplication event leading to TNKS1 and TNKS2 occurred at some point during the evolution of fishes [53]. Our analysis shows that the two TNKS paralogs can be identified in representatives of bony fishes (Osteichthyes) and cartilaginous fishes (Chondrichthyes), but not in the jawless fishes *Petromyzon marinus* and *Eptatretus burgeri*, indicating that the duplication event took place during the early evolution of vertebrates.

To interrogate a possible pre-metazoan origin, we searched for the presence of TNKSs in representative subsets of earlier diverging eukaryotes. No TNKSs were identified in the major eukaryotic groups Archaeplastida and Holomycota, which include plants and fungi, respectively. We then searched for the presence of TNKSs in groups more closely related to metazoans. While we did not find TNKSs in Filasterea or Ichthyosporea, we identified sequences coding for TNKSs in transcriptome shotgun assembly (TSA) data from the choanoflagellates *Salpingoeca kvevrii* and *Salpingoeca macrocollata*. Choanoflagellates are the proposed sister group of metazoans and are studied as models for pre-metazoan evolution along with their ability to live in communities and differentiate into different cell types [57,58]. It is worth mentioning that we did not detect the presence of TNKSs in the model choanoflagellates *Monosiga brevicollis* and *Salpingoeca rosetta*, for which the genomes have been sequenced [58,59]. However, these model choanoflagellates are relatively closely related and do not sufficiently represent the genetic diversity found among choanoflagellates [60]. A phylogenetic tree constructed from the TNKS protein sequences of several metazoan species as well as the choanoflagellates *S. kvevrii* and *S. macrocollata* reproduced the expected phylogenetic relationships (Figure 2b and Appendix A). These results suggest an origin of TNKSs in choanoflagellates.

Although several proteins such as Shank1, ANKS1b and ASZ1 are known to comprise combinations of ANK and SAM domains (Figure 2c), this domain architecture appears unique to TNKSs among PARPs. We speculated that gene-fusion or exon-shuffling [61] between an ancestral PARP and an ANK-SAM protein could represent obvious mechanisms by which TNKSs may have originated. To see if tangible TNKS precursor proteins could be identified, we searched with sub-domain sequences from human TNKS1 against non-metazoan sequence databases. Surprisingly, we identified proteins with ANK domains that share 40–50% sequence identity with the ARC5 domain of TNKS1 in diverse phylogenetic groups including prokaryotes. However, the high similarity may be explained by the fact that many residues in ANK-folds are well-conserved irrespective of the domain’s function [62]. Indeed, closer inspection of the hit proteins such as the ANK domain from the archaeon *Acidianus manzaensis* (Figure 2c) showed that binding of TBM peptides in a similar mode as observed for TNKS ARC domains would not be possible (Appendix A). From a similar search using the TNKS1 ART domain as the query, we identified a yet uncharacterized protein from the pathogenic amoeba *Naegleria fowleri* that shares 37% sequence identity with the ART domain of TNKS1 (Figure 2c). Further analysis indicates that the TNKS-like ART domain from *N. fowleri* (*Nf*TNKSL) contains a putative zinc-binding motif corresponding to the motif in the ART domain of TNKSs (Appendix A). It was thought that this motif was unique to TNKSs among PARPs [63], and we showed recently that it is important for the structural integrity of the TNKS ART domain [44]. The available sequence information of *Nf*TNKSL indicates that it does not contain other domains. Considering the similarity, *Nf*TNKSL likely shares a common ancestor with TNKSs and it may possibly resemble an ancestral version of TNKSs before acquisition of ANK and SAM domains.

### 3.2. Conservation of Tankyrase Sequence and Structure

The results above confirm the vast distribution of TNKSs among metazoan groups and additionally in choanoflagellates. We next aimed to characterize how well-conserved TNKSs are throughout evolution in terms of sequence, structure and function. The following analyses were performed in respect to metazoans with well-characterized genomes; choanoflagellates were omitted from the analyses as their sequence information originated from TSA and may thus be incomplete and less reliable.

First, we compared the sequence identity of human TNKS1/2 and other human ARTD family members with their orthologs from representative metazoan species as derived from multiple sequence alignments (Figure 3). For further comparison, the ubiquitous and well-conserved proteins cytochrome C [64] and seryl-tRNA synthetase [65] were included as controls. It was reported that some PARPs such as PARP4 and PARP9 have experienced rapid evolution likely due to positive selection in response to host–virus conflicts [66]. Indeed, PARP4, PARP9 and PARP12 share less than 50% sequence identity between the homologs of humans and the fish *S. salar*. In contrast, TNKSs showed the highest degree of conservation of any analyzed family member, displaying approximately 60% sequence identity between human TNKS1/2 and the distant TNKS ortholog found in the demosponge *A. queenslandica*. The degree of conservation appears comparable to that of cytochrome C and seryl-tRNA-synthetase, suggesting that TNKSs are highly conserved members of the ARTD family. Remarkably, the level of conservation for TNKSs within the metazoan lineage appears greater than that for PARP1, which is known to act as key-player in DNA damage response [67] and its orthologs are widely distributed throughout eukaryotes in protists, animals, fungi and plants [53].

To gain insight into the localization of potentially highly or poorly conserved regions in TNKSs, we plotted the number of different residues for each position from an alignment of 113 TNKS sequences from taxonomically diverse metazoans (Figure 4a). We additionally performed analysis by ConSurf, which provides normalized grades for the conservation at each position [41]. Because the N-terminal region in TNKS1 is poorly conserved among species, we used human TNKS2 as reference protein to avoid ambiguity due to insufficient alignments in this region.

This analysis shows strong conservation for the ARC domains 1–2 and 4–5, the SAM domain and the ART domain. The ARC3 domain clearly shows a lower degree of conservation in comparison to the other ARCs. ARC3 was reported to not bind TBMs and likely has a structural role [14,15,69]. Overall, the lowest degree of conservation is found in the regions linking the domains ARC3 to ARC4, ARC5 to SAM and SAM to ART as well as the N- and C-terminal regions. These regions are likely intrinsically disordered, thus allowing a higher freedom in residue composition. A study by Eisemann et al. investigated the structural features of the ARC region from TNKS1 and demonstrated that the linking region from ARC3 to ARC4 is indeed flexible, while the regions from ARC1 to ARC2 and ARC2 to ARC3 form rigid interfaces [14]. Although points of flexibility were also observed between ARC4 and ARC5 [14], our analysis shows a high level of conservation at their interface, possibly indicating that the flexibility does not originate from an intrinsically disordered linker region.

In order to further characterize the sequence conservation of TNKSs in respect to their structures, we mapped strongly conserved residues to TNKS domain structures. From an alignment of the TNKS sequences from 11 metazoan species with well-characterized genomic information (Figure 4b), we mapped only identical residues to the domain structures of ARCs, SAM and ART (Figure 4c–e and Appendix A). The ARC domains 1, 2, 4 and 5 show a large, continuous patch of highly conserved residues around the TBM-binding pocket (Figure 4c and Appendix A), indicating that the ability to bind the same peptide motif is conserved throughout TNKS evolution. Such a conserved patch is missing for ARC3 (Appendix A), which is in agreement with its inability to bind TBM peptides in human TNKSs. Indeed, inspection of ARC3 sequences from early diverging metazoans confirms the absence of crucial residues required for binding TBMs (Appendix A), suggesting that the ARC3 domain did not only recently lose the TBM-binding function. Sequences from the SAM domains have fewer conserved sites in common (Figure 4d); however, the residues at its interaction sites appear highly conserved. These interaction sites are termed “mid-loop” and “end-helix” and allow the TNKS SAM domains to multimerize via head-to-tail interactions [18,19,20]. The ART domain shows a high degree of conservation around the NAD^+^-binding site (Figure 4e). The binding mode of NAD^+^ is therefore likely the same across TNKSs from different species. The residues of the donor loop (D-loop) are likewise entirely conserved. The D-loop was shown to adapt different conformations and it partially occupies the NAD^+^-binding pocket in crystal structures of apo-TNKSs [1]; it was speculated to be involved in the activity regulation of TNKSs [70]. Overall, these results suggest a remarkable level of conservation among TNKSs and imply that the molecular functions across TNKSs are likely conserved.

### 3.3. The Basic Molecular Characteristics of TNKSs Are Conserved

The human paralogs TNKS1 and TNKS2 are the most studied TNKS proteins. As shown above, the sequences of TNKSs across all metazoans are highly conserved. While the distant TNKS homolog in drosophila was functionally characterized through in vivo studies in terms of cellular function [31,71,72,73], we wanted to confirm that the molecular functions of distantly related TNKS homologs are conserved using in vitro studies with purified proteins. For this, we recombinantly produced TNKS proteins of a basal metazoan, the demosponge *A. queenslandica* (*Aq*TNKS), comprising the ART domain, the ART-SAM domains and the ARC5 domain. Corresponding constructs from human TNKS1 were tested for comparison. First, we tested the poly-ADP-ribosylation activity of the ART and SAM-ART constructs by auto-modification (Figure 5a). The proteins were incubated with biotinylated NAD^+^ and the incorporation of biotin into the PAR chains was detected in Western blot using HRP conjugated streptavidin. A smear corresponding to the PARylation of the proteins was clearly detected for the SAM-ART constructs of *Aq*TNKS and TNKS1, but not for the isolated ART domains. Moreover, to detect PAR chains that originate from PARylation activity during the recombinant expression, we used the PAR-binder ALC1 fused to nanoluciferase [49]. Similarly, PAR chains were detected only for the SAM-ART constructs, and incubation with PAR hydrolyzing human PARG resulted in loss of the smears. These results are in agreement with previous reports showing a significant loss of catalytic activity of human TNKS1/2 in absence of the SAM domain [7,19,20,21,70]. Therefore, the role as poly-ADP-ribosyltransferases and the SAM-dependent activation mechanism are likely conserved among TNKSs. Moreover, size-exclusion experiments with *Aq*TNKS constructs showed formation of multimeric species for the SAM-ART protein but not the ART domain, which is in agreement with the reported multimerization function of the SAM domain in human TNKSs (Appendix A) [18,19,20,21].

To further demonstrate the high similarity between the ART domains of *Aq*TNKS and TNKS1, we tested the thermal stabilization with established potent and selective TNKS inhibitors that were recently benchmarked [50]. This included adenosine-site binders IWR-1, G007-LK, OM-153 and compound 40 [74,75,76,77], nicotinamide-site binders XAV939, E7449, AZ6102 [10,78,79] and the dual-site binder NVP-TNKS656 [80]. For all inhibitors, the melting temperatures for both TNKS1 and *Aq*TNKS were similarly increased (Figure 5b), underlining the structural similarity between these ART domains.

Next, we tested the binding of a canonical TBM peptide to the ARC5 domain constructs of *Aq*TNKS and TNKS1. Guettler et al. developed an optimized peptide (sequence: REAGDGEE) that possesses high binding affinity to the ARC domains of human TNKSs [15,43]. We expressed the ARC5 domain constructs and the TBM peptide as fusion proteins with the fluorescent proteins CFP and YFP, respectively. To study the interaction of the ARC5 constructs with the TBM peptide, we measured FRET emissions upon mixing of the constructs (Figure 5c). Both ARC5 constructs from TNKS1 and *Aq*TNKS interact with the TBM fused to YFP with similar binding affinities, while they showed no interaction with YFP lacking the TBM. In combination with the high conservation of the residues lining the ARC5 domain, these results suggest that the ability of TNKSs to bind the same TBM sequences is highly conserved.

### 3.4. Acquisition of Tankyrase Binding Partners throughout Evolution

Experimental data supporting *bona fide* interaction for many of the reported TNKS binding partners in humans exists, although only limited information is available for the interaction of proteins with TNKS orthologs in other metazoans [31]. Considering the strong conservation of TNKSs and the ability of *Aq*TNKS to bind a peptide with the canonical TBM, it is reasonable to think that TNKSs throughout the metazoan lineage can bind this motif. The conservation of the TBMs in orthologs from binding partners of human TNKSs has been reported for selected binders, covering often only a very limited taxonomic breadth [10,23,24,25].

To gain insight into the conservation of TNKS binding partners, we selected a set of 21 experimentally confirmed TNKS interaction partners (Table 2) and determined the presence of TBMs in the orthologs of several metazoan species (Figure 6). Only motifs were considered that aligned with the position of TBMs from the human orthologs. In addition to the canonical TBM Rxx[ACGP]xGxx, we also marked the presence of the less constrained motifs Rx(4)Gxx and Rx(5)Gxx, which were reported for some TNKS interaction partners in humans [10,16,17], although the target may require multiple of such motifs to achieve sufficient avidity.

Overall, the analysis revealed that the conservation of corresponding TBMs compared to the human orthologs varies greatly for different proteins. A few of the analyzed orthologs seem to have acquired the TBMs relatively recently, such as GDP-mannose 4,6-dehydratase (GMD) or the telomere length regulator TRF1. Both do not harbor corresponding TBMs in orthologs of mice or in any of the earlier-diverging metazoans, with the exception of a possible TBM in the GMD ortholog from *P. marinus*. Interestingly, GMD is an otherwise highly conserved enzyme important in basic metabolic pathways, and orthologs of GMD are present in prokaryotes and eukaryotes [81]. While it was reported that GMD is not a direct target of PARylation by TNKSs and moreover appears to inhibit the catalytic activity of TNKSs [24], the exact function of the GMD-TNKS interaction is not well-understood at this point. In contrast, the interaction of TNKSs with Axin1 (Axin) appears highly conserved, as possible TBMs were found in orthologs from all species used for the analysis. Axin is a crucial component in the Wnt signaling pathway and interacts in humans with two TBMs with TNKSs; however, we omitted the non-canonical second motif from the analysis as the binding rules for it are not clear [16]. Earlier work by Feng et al. also confirmed interaction of TNKS and Axin in *D. melanogaster* [31]. TBMs in Golgin-45 were identified in orthologs from many of the analyzed species with exception of *A. queenslandica*. Golgin-45 is a Golgi-associated protein required for Golgi structure and protein transport. Zhang et al. identified Golgin-45 as a target for PARylation by TNKSs [82]. More recently, it was also shown to be responsible for TNKS1 localization to the Golgi apparatus during interphase [83].

**Table 2 biomolecules-12-01688-t002:** Overview of tankyrase interaction partners used for the analysis. The required arginine and glycine in each TBM is shown in bold and underlined. The residue number of the arginine in the first position of the TBM from the human ortholog is shown.

TNKS Interaction Partner	Role or Cellular Function	TBM in Human Ortholog	Reference
AMOT	Hippo/YAP signaling	77-**R**QEPQ**G**QE	[11]
Arpin	Cell migration	213-**R**EQGD**G**AE	[84,85]
ATG9A	Pexophagy	233-**R**LPGL**G**EA	[26]
Axin1 *	Wnt signaling	22-**R**PPVP**G**EE	[10,14,16,31]
CASC3	Pre-mRNA splicing	146-**R**QSGD**G**QE	[15,82]
Dicer1	RNA interference	656-**R**ELPD**G**TF	[86]
GMD	GDP-mannose 4,6 dehydratase	12-**R**GSGD**G**EM	[24,87]
Golgin-45	Golgi structure and protein maturation	18-**R**GAGD**G**ME	[15,82,83]
IRAP	Vesicle trafficking	96-**R**QSPD**G**AC	[5,6,14,23]
Mcl-1	Apoptosis regulation	78-**R**PPPI**G**AE	[15,88]
MERIT40	DNA damage repair	28-**R**SNPE**G**AE 48-**R**SEGE**G**EA	[15,89]
NKD2	Wnt signaling	16-**R**ESPE**G**DS	[86]
Notch2	Notch signaling	1726-**R**REPV**G**QD	[86]
NuMA	Mitotic spindle assembly	1743-**R**TQPD**G**TS	[15,23,90]
PEX14	Peroxisome homeostasis	310-**R**MEVQ**G**EE 350-**R**RGGD**G**QI	[26]
SH3BP2	Osteoclast formation	415-**R**SPPD**G**QS	[7,15]
SH3BP5	Epithelial lumen formation	269-**R**GCGV**G**AE 368-**R**SECS**G**AS	[91]
SOX9	Chondrocyte differentiation	257-**R**PLPE**G**GR 271-**R**DVDI**G**EL	[92]
Striatin	Dendritic Ca^2+^ signaling	302-**R**SAGD**G**TD	[15,93]
TAB182	Cytoskeletal maintenance	1508-**R**PQPD**G**EA	[15,23]
TRF1	Telomere maintenance	13-**R**GCAD**G**RD	[2,23,87]
USP25	Ubiquitin protease	1049-**R**TPAD**G**R	[15,25]

* The second non-canonical TBM from Axin1 is not shown here.

In our view, TNKSs are not likely to bind the orthologs that show an absence of corresponding TBMs, although it cannot be ruled out that some of them may still interact with TNKSs, for example due to the presence of non-canonical binding motifs or because of TBMs situated in different positions compared to the binding partner in human. Moreover, it is striking that many of the human TNKS binding partners appear to entirely lack corresponding orthologs in several species. For example, we could not identify orthologs of TAB182 in *X. laevis* or any earlier-diverging metazoans. Orthologs of AMOT are present in all analyzed vertebrates, while no orthologs could be identified in any of the invertebrates. Notably, all identified orthologs of AMOT possess a TBM corresponding to the human ortholog, indicating that an interaction with TNKSs may have evolved not long after the emergence of AMOT.

Due to the many TBMs present in the C-terminus of the prominent TNKS binding partner RNF146 [17], a different analysis was performed for this protein (Appendix A). RNF146 is an E3 ubiquitin ligase, and it is associated with PAR-dependent proteasomal degradation of TNKS target proteins [82,94,95,96,97]. All possible TBMs were mapped to the sequences of RNF146 orthologs, showing that many of these proteins possess possible TBMs in different numbers, configurations and positions compared to human RNF146. While it is difficult to predict which of these orthologs may interact with TNKSs, a recent study showed that in *D. melanogaster*, TNKS PARylation-dependent proteasomal degradation is mediated by RNF146 [98], although a direct interaction with TNKSs was not investigated.

The results obtained here lead to the overall picture that TNKSs themselves are highly conserved, while the interactions with binding partners seem to be far more variable throughout evolution as suggested in part by the varying presence of TBMs in orthologs and also because several of the binding partner orthologs appear to be simply not present in many species.

## 4. Discussion

### 4.1. Origin and Distribution of Tankyrases

The evolutionary transition from a single-celled urchoanozoan (ancestor of choanoflagellates and metazoans) to metazoans occurred together with the innovation and loss of gene families [60,99]. Proteins that originated during this transition and that were retained in the metazoan lineage are of interest for investigations about the evolutionary origin of metazoans, as these proteins may have been important factors for allowing the development of multicellular animals.

TNKSs seem to appear only sparsely in choanoflagellates but are vastly distributed throughout metazoans, which suggests that TNKSs may be important for multicellular development. However, the presence of TNKSs seems not to be obligatory for metazoan life, as evidenced by the reported loss of TNKS in *C. elegans* [52,53] and possibly in placozoans as shown in this work.

As part of the analysis, we identified a yet uncharacterized ARTD family member in the pathogenic amoeba *N. fowleri*, here termed *Nf*TNKSL. Based on the sequence information available, *Nf*TNKSL contains only a single ART domain, which shows more similarity to the ART domain of TNKSs than that of any other ARTD family member. It is tempting to speculate that *Nf*TNKSL may resemble an ancestral TNKS precursor, which later obtained ANK and SAM domains through domain-shuffling, a common mechanism for the innovation of novel protein functions in the stem animal lineage [100,101,102].

### 4.2. Conservation of Tankyrases

We have shown that the sequences of TNKSs are highly conserved throughout metazoans, displaying similar levels of conservation to fundamentally important proteins such as cytochrome C. As it was expected, the degree of conservation is not equal throughout the TNKS sequence but corresponds to the structure of the protein. For example, regions thought to be flexible linkers appear less conserved, although some of them are likely important for the function of TNKSs [14].

Mapping of conserved regions to the TNKS structures showed a very high degree of conservation particularly around the TBM binding site of ARCs and the NAD^+^ binding site of the ART domain. Functional in vitro studies comparing human TNKS1 with *Aq*TNKS from *A. queenslandica,* a sponge that diverged from other metazoans over 600 million years ago [103], confirmed that the basic molecular functions are conserved. We demonstrated that *Aq*TNKS is a poly-ADP-ribosyltransferase that has strongly increased activity in the presence of the SAM domain, and that the ARC5 domain binds a canonical TBM with similar affinity compared to human TNKS1. The ability of *Aq*TNKS to bind structurally diverse TNKS inhibitors that display high selectivity and nanomolar binding affinity against human TNKSs further highlights the high degree of conservation.

### 4.3. Acquisition of Tankyrase Binding Partners

From an analysis of the presence of TBMs in orthologs of human TNKS binding partners, the number of orthologs with corresponding TBMs decreased according to the species’ divergence from humans. Our analysis indicates that new TBMs formed throughout different animal lineages. This may leave the impression of a trend favoring accumulation of TNKS binding partners as the complexity of animal species increases; however, it is important to consider that the loss of TBMs may have occurred as well. The vast majority of TNKS binding partners were only experimentally confirmed in humans, which limits our analysis to a possible emergence of TBMs in orthologs of these binding partners; a comprehensive analysis for the loss in TBMs would require multiple experimentally verified TNKS binding partners in early diverging metazoan lineages. From the current results, we cannot exclude the possibility that the number of TNKS interaction partners for example in early diverging groups such as sponges or arthropods may be similar to the number found in humans.

Hub proteins that bind small linear motifs (SLiMs) were recently suggested to be grouped under the term of “linear motif-binding hubs” (LMB-hubs) [104]. With TBMs being SLiMs, and considering the functions of TNKSs, they certainly can be classified as LMB-hubs. Evolutionary analyses previously performed on other LMB-hub proteins established striking parallels also observed here for TNKSs: While the hub proteins are generally highly conserved, the presence of interacting SLiMs is often poorly retained in the orthologs of binding partners [105,106]. This may be the case because SLiMs can rapidly evolve de novo or similarly be lost again through small changes in amino acid sequences [107]. SLiMs are located on intrinsically disordered protein regions such as the protein termini, which are often far less mutationally constrained than structured regions—allowing rapid changes in sequence composition [108]. Furthermore, SLiM-hub interactions may allow a level of fine-adjustment for the regulation of binding partners through change of the SLiM sequence, resulting in changes in binding affinity to the hub protein [107].

Indeed, it has been reported that differences in the sequence of TBMs result in varied binding affinities to TNKSs [15,87]. A study by Eisemann et al. recently demonstrated that mutations in the TBM of TRF1 reduced the binding affinity and also resulted in lowered PARylation of TRF1 by TNKS1 in vitro. On an evolutionary level, the TNKS-TBM interaction may allow for TNKS binding partners to adapt their TBMs and thereby adjust the degree by which they are targeted by TNKS-mediated PARylation. This adjustment of binding may not only happen through changes in the TBM sequences but could also take place through evolution of additional TBMs in the TNKS binding partners or via their dimerization, which creates a complex relationship in terms of multivalent binding to the multimeric TNKS scaffolds.

### 4.4. Implications for the Development of Therapeutics Targeting Tankyrases

The cellular or physiological functions of TNKSs and the biological outcomes thereof can be attributed to the diverse assortment of binding partners from many different pathways that are regulated by TNKSs. While the structure and molecular functions were assessed to be highly conserved among TNKSs, the cellular functions of TNKSs may vary greatly depending on the species as different sets of TNKS binding partners may be regulated by TNKSs. For example, in *D. melanogaster*, we could identify only 11 orthologs of the 21 analyzed human TNKS binding partners, and only 4 of these orthologs harbor corresponding TBMs. Another example can be found in TRF1, a nuclear protein repressing telomere elongation which interacts through a TBM with TNKSs in humans. While it was reported that TNKSs have an important role in telomere maintenance through regulation of TRF1 in humans [2,3,4,109,110,111], the TBM is absent from the TRF1 ortholog in mice, where TNKSs were shown not to participate in the regulation of telomere length in multiple studies [28,112,113]. Although the attenuation of telomere elongation through inhibition of TNKSs could be a potential avenue for cancer therapeutics [114], mouse models would not adequately represent the potential efficacy of such drugs. On the other hand, inhibition of TNKSs was shown to reduce Wnt signaling activity through blocking the PARylation of Axin [10], which has a highly conserved TBM across species. Treatment with TNKS inhibitors leads to reduced proliferation of cancer cells in Wnt-dependent tumors [115] and mice are frequently used as model organisms in preclinical studies.

For the evaluation of TNKS inhibitors as therapeutic drugs, it should be considered that the cellular functions of TNKSs may differ greatly in humans and model organisms, which may limit for example how well the perceived efficacy or toxicity translates to humans.

## 5. Conclusions

The main points derived from this work are the following: First, we showed that TNKSs are distributed widely throughout metazoans, although they seem to have a pre-metazoan origin as indicated by their presence in some choanoflagellates. Second, our analyses indicate that TNKSs are remarkably conserved in terms of sequence, structure and molecular function. Last, the conservation of TBMs in TNKS binding partners and their orthologs appears to vary greatly, indicating a dynamic gain and possibly a loss of TBMs throughout evolution. Moreover, many orthologs of human TNKS binding partners appear to be missing in several of the species analyzed, which further contributes to the notion that TNKSs are likely regulating different sets of processes in different species.

## Figures and Tables

**Figure 1 biomolecules-12-01688-f001:**
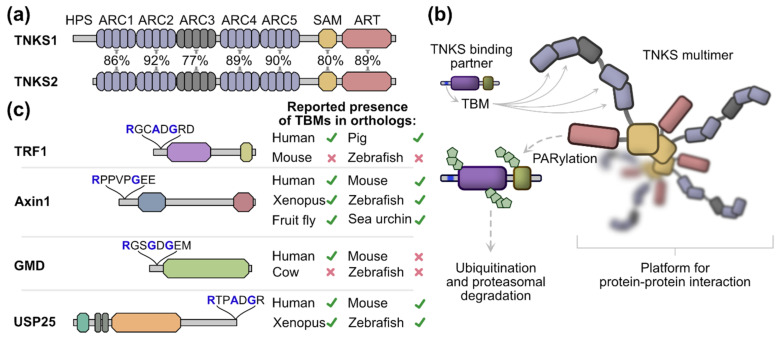
Tankyrase domain organization, function and binding partners. (**a**) Schematic representation of the domain architecture from human TNKS1 and TNKS2. The sequence identities between domains of TNKS1 and TNKS2 are shown in percentages. TNKS1 has a Histidine, Proline and Serine-rich region (HPS) at the N-terminus, which is not present in TNKS2. (**b**) A model of TNKS function. TNKSs form multimers mediated by the SAM domain and interact with proteins that contain TNKS binding motifs (TBMs), which bind to the ARC1, 2, 4 and 5 domains of TNKSs. Many binding partners are PARylated by TNKSs, which leads to their subsequent ubiquitination and proteasomal degradation. Additionally, TNKSs may also act as scaffolding proteins by providing a platform for protein–protein interactions. (**c**) Examples of tankyrase binding partners TRF1, Axin1, GMD and USP25. The TBM sequence for each binding partner is shown. Crucial residues for the interaction with TNKS ARC domains are highlighted in blue. Previously reported presence or absence of the TBM in orthologs from other species is shown for TRF1 [23], Axin1 [10], GMD [24] and USP25 [25].

**Figure 2 biomolecules-12-01688-f002:**
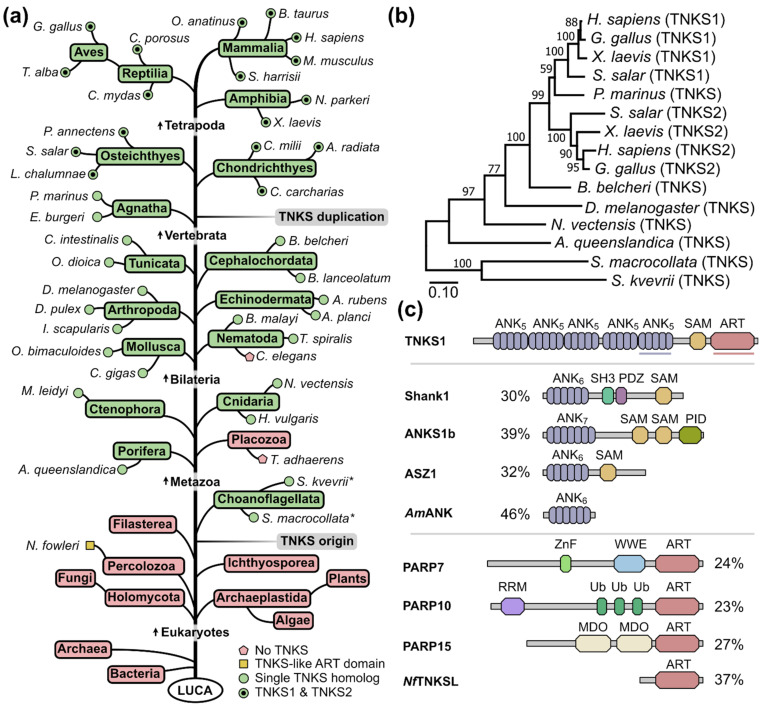
Distribution and possible origin of tankyrases. (**a**) The presence or absence of TNKSs in phylogenetic groups was determined from searches of various protein and genomic databases. Phylogenetic relationships among groups are schematically depicted. Representative species are shown for metazoans and selected groups and the presence of a single TNKS or both TNKS1/2 paralogs is indicated. Information with common names for the phylogenetic groups and species displayed here together with identifiers for the protein sequences can be found in Appendix A. Species in which TNKSs were identified only from TSA data are marked (*). LUCA = Last universal common ancestor. (**b**) Phylogenetic reconstruction from TNKS sequences using the maximum likelihood method. Percentages for bootstrap support values are shown for each node. Bootstrap tests were conducted with 1000 bootstrap replicates. The analysis was performed with truncated TNKS sequences starting from the ARC3 domain to match the likely incomplete sequence from *S. macrocollata*. In Appendix A, the reconstruction was alternatively performed with the neighbor joining method. (**c**) Schematic representation of proteins with domain architectures related to TNKSs. Sequence identity percentages to the ARC5 or ART domain of human TNKS1 were derived from local sequence alignments. Shank1, ANKS1b and ASZ1 are human proteins that contain both ANK and SAM domains. The ANK protein from *Acidianus manzaensis* (*Am*ANK) shares a high sequence identity with ARC5 of TNKS1. The number of ankyrin repeat units is indicated for each ANK domain. PARP7, PARP10 and PARP15 are human representatives from the clade 3 of the ARTD family, which is most closely related to clade 4 (TNKSs) [52]. The TNKS-like protein from the amoeba *N. fowleri* (*Nf*TNKSL) is described in more detail in Appendix A.

**Figure 3 biomolecules-12-01688-f003:**
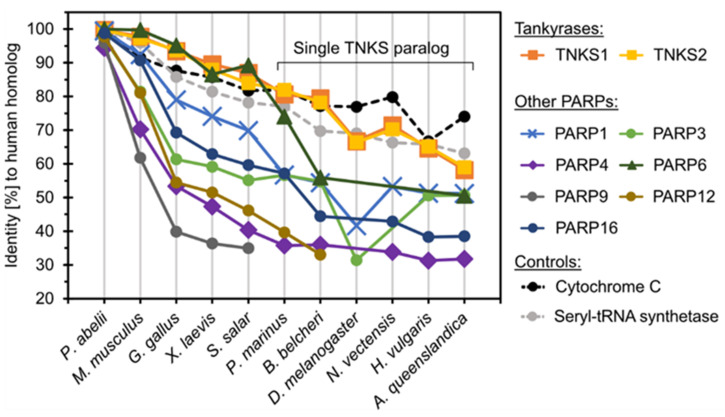
Sequence conservation of tankyrases and representative PARP proteins. For the different proteins analyzed, the graph shows the sequence identity shared by the human ortholog with orthologs from different metazoan species. The identities were determined from multiple sequence alignments of the orthologs from each set of ARTD family members. In species that only encode a single TNKS paralog, the sequence identity shared with human TNKS1 or TNKS2 was determined against the same TNKS ortholog. The conserved proteins cytochrome C and seryl-tRNA synthase were used for comparison.

**Figure 4 biomolecules-12-01688-f004:**
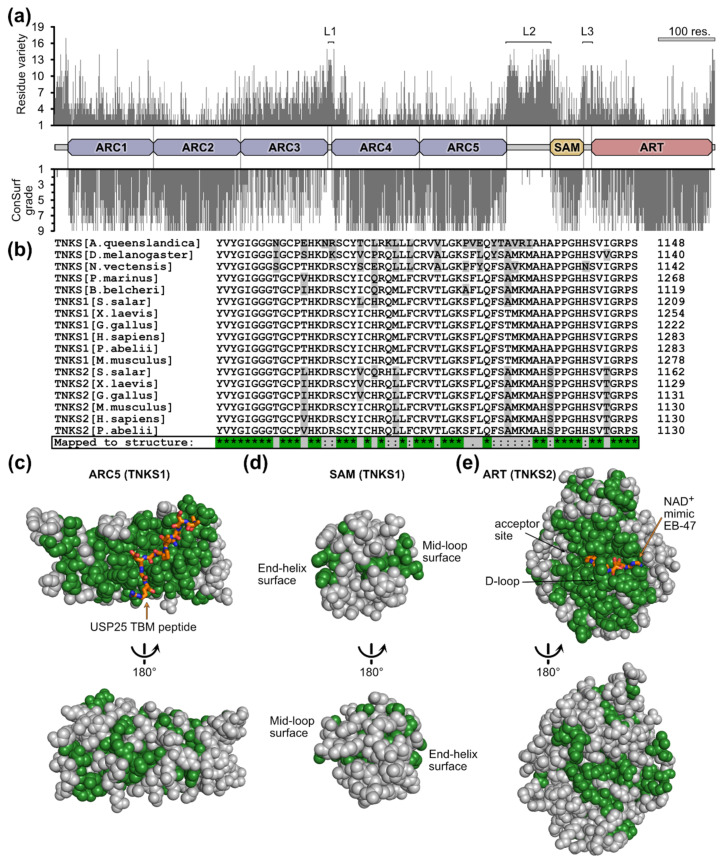
Conservation of tankyrase residues linked to the protein structure. (**a**) Residue-specific conservation plot. The number of different residue identities and ConSurf grade was plotted for each residue from an alignment of 113 TNKS sequences. The ConSurf grade is a normalized measure for the conservation and ranges from 1 (low conservation) to 9 (high conservation). The three likely flexible linker regions are designated L1, L2 and L3. The projected domain boundaries were assigned based on existing structural information of individual domains from human TNKS1 and TNKS2. A scale corresponding to 100 amino acid residues is shown. (**b**) A representative section of a multiple sequence alignment from the TNKS sequences of 11 metazoan species. This section covers a part of the ART domain sequence. Residues highlighted in gray are different compared to human TNKS1. Only residues identical in every TNKS sequence were mapped to the structures and are shown in green: (**c**) Structure of the ARC5 domain from human TNKS1 in complex with the TBM peptide from USP25 (orange) (PDB: 5GP7, [25]). (**d**) Structure of the SAM domain from human TNKS1 (PDB: 5KNI, [20]). (**e**) Structure of the ART domain from human TNKS2 in complex with the NAD^+^-mimicking compound EB-47 (orange) (PDB: 4BJ9, [68]).

**Figure 5 biomolecules-12-01688-f005:**
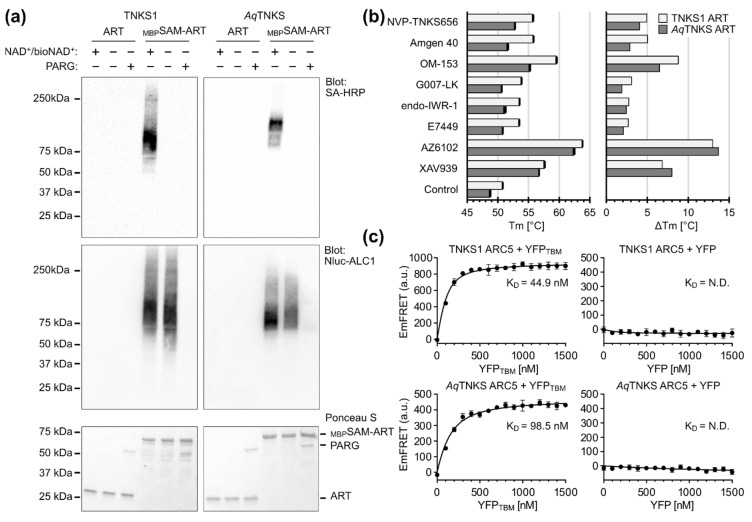
Experimental validation of the tankyrase molecular functions from *Amphimedon queenslandica*. (**a**) Western blot of ART and SAM-ART constructs from human TNKS1 and *A. queenslandica* TNKS (*Aq*TNKS). SAM-ART constructs contain an N-terminal maltose-binding protein (MBP) tag. TNKS constructs (2 μM) were incubated for 1 h at room temperature in presence or absence of a mixture containing biotin-NAD^+^/NAD^+^ (1:10). The total NAD^+^ concentration was 2 μM for TNKS1 and 10 μM for *Aq*TNKS due to a lower activity observed in *Aq*TNKS. PAR with incorporated biotin was detected with streptavidin conjugated to horseradish peroxidase (SA-HRP), while total PAR modification independent of biotin was detected using nanoluciferase fused to ALC1 (Nluc-ALC1). PARG (0.2 mg/mL) was added to verify a removal of PAR chains. (**b**) Thermal stabilization of TNKS ART domain constructs by selective TNKS inhibitors. Differential scanning fluorimetry (DSF) was performed after mixing TNKS ART domain constructs (5 μM) and TNKS inhibitors (25 μM). A control without inhibitor was included. The absolute melting temperatures (Tm) are shown on the left and data shown are mean ± standard deviation with number of replicates *n* = 4; the melting temperature relative to the respective controls calculated from the mean values are shown on the right. (**c**) FRET-based determination of the dissociation constant K_D_ for ARC5 constructs with a TBM peptide. CFP-fused ARC5 constructs (100 nM) from TNKS1 or *Aq*TNKS were mixed with different concentrations of YFP fused to the TBM peptide (REAGDGEE). As control, the CFP-ARC5 constructs were mixed with YFP without the TBM peptide. The fluorescence emissions from FRET (EmFRET) were determined by a method described by Song et al. [51]. Data shown are mean ± standard deviation with number of replicates *n* = 4.

**Figure 6 biomolecules-12-01688-f006:**
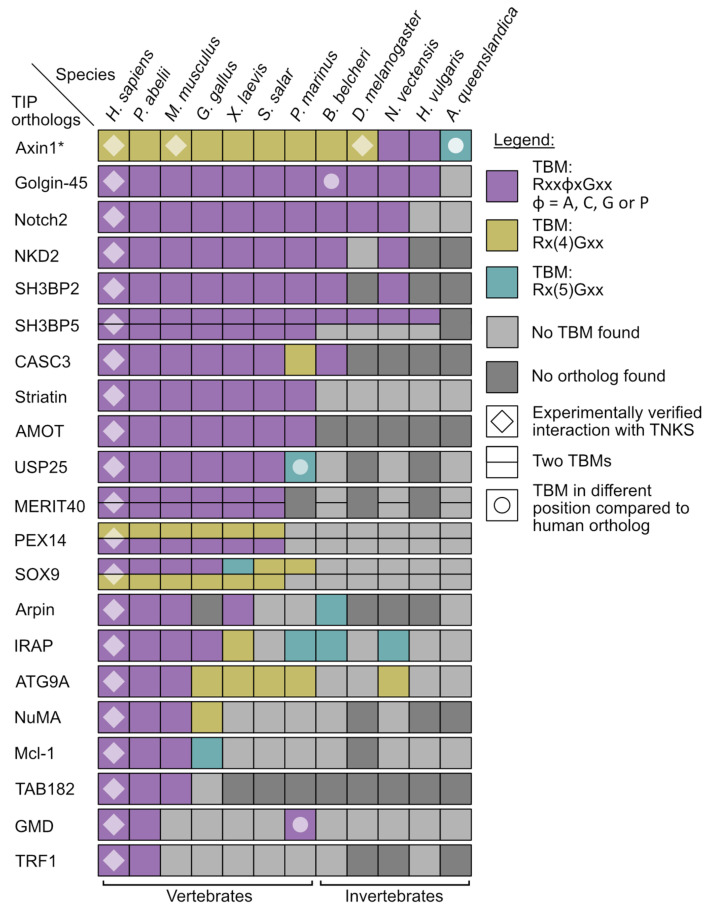
Identification of corresponding tankyrase binding motifs in orthologs of confirmed interaction partners. Orthologs of 21 TNKS interaction partners (TIPs) were examined for presence or absence of possible TBM sequences. Three configurations Rxx[ACGP]xGxx, Rx(4)Gxx and Rx(5)Gxx were considered as TBM. TBMs were only considered if the motif aligned with the TBM in the human ortholog in a multiple sequence alignment, although in some cases also TBMs that did not align but were found in nearby positions are shown (see legend). * The non-canonical second motif from Axin1 was not included in this analysis.

**Table 1 biomolecules-12-01688-t001:** Species frequently referred to in this work.

Species	Common Name	Group	Notes
*Homo sapiens*	Modern human	Apes	Most studies of TNKS and its interaction partners were carried out in human model systems.
*Pongo abelii*	Sumatran orangutan	Apes	Extant ape species that is phylogenetically most distant from human.
*Mus musculus*	House mouse	Mammals	Most widely used mammalian model organism.
*Gallus gallus*	Red junglefowl	Birds	Includes the domesticated chicken.
*Xenopus laevis*	African clawed frog	Amphibians	Model amphibian.
*Salmo salar*	Atlantic salmon	Bony fishes	Representative of fishes with high-quality genome sequence data.
*Petromyzon marinus*	Sea lamprey	Jawless fishes	Model for vertebrate evolution. An early diverging vertebrate with only one TNKS instead of two paralogs (TNKS1/2).
*Branchiostoma belcheri*	Belcher’s lancelet	Lancelets	Closely related to vertebrates. Model organism for vertebrate evolution.
*Drosophila melanogaster*	Fruit fly	Insects	Popular model organism in genetics and developmental biology.
*Caenorhabiditis elegans*	*C. elegans*	Roundworms	Popular model organism for studying neuronal development. Gene-loss of TNKS and other PARPs was reported.
*Nematostella vectensis*	Starlet sea anemone	Cnidarians	Model cnidarian.
*Hydra vulgaris*	Fresh-water polyp	Cnidarians	Model organism in regenerative biology.
*Amphimedon queenslandica*	N/A	Sponges	Model for metazoan evolution and developmental biology. Functions from its TNKS ortholog were experimentally characterized in this work.
*Salpingoeca kvevrii*	N/A	Choanoflagellates	The most diverged TNKS orthologs identified in this work are from these choanoflagellates.
*Salpingoeca macrocollata*			
*Naegleria fowleri*	Brain-eating amoeba	Percolozoa	Non-metazoan eukaryote that encodes an ART domain with high similarity to that of TNKSs.

## Data Availability

The data presented in this study are available in manuscript and in the Appendix A.

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
