# Peer review of "An Evolutionary Perspective on the Origin, Conservation and Binding Partner Acquisition of Tankyrases"

_biomolecules, 2022, doi:10.3390/biom12111688_

Round 1
Reviewer 1 Report
This is a very interesting article on the evolutionary implications of tankyrase proteins which are PARP-domain-containing proteins that regulate the activities of a wide repertoire of target proteins via post-translational PARylation. Expansive roles of tankyrase activity have recently emerged in the development and maintenance of the proteome, and impact telomere elongation, DNA repair, metabolism, and apoptosis. This paper traces the distribution of tankyrases throughout the branches of multicellular animals and identifies the single-celled choanoflagellates as a potential early origin of tankyrase proteins. The authors further characterized an anciently diverged conserved tankyrase homolog from the sponge, and emphasized that although catalytic function is conserved, interaction partners may be highly variable between species, as a potential etiology of their diverse repertoire of PARylated and non-PARylated protein targets.
The article is clear and well-written and the sequence homology analyses of ARC and SAM domains of the 113 tankyrase proteins, as well as the experimental functional validations are well done. The discussion is clear, brief, and interesting, These sequence analyses will be a useful contribution to our general understanding of proteostasis and molecular evolution. The article should be published without delay, and perhaps the discussion may be strengthened about if some hypotheses could be elaborated on how this conserved sequence information might inform us on how Tankyrases might exert control over multiple signaling pathways through both PARP-dependent and PARP-independent mechanisms.
Author Response
Reviewer #1
This is a very interesting article on the evolutionary implications of tankyrase proteins which are PARP-domain-containing proteins that regulate the activities of a wide repertoire of target proteins via post-translational PARylation. Expansive roles of tankyrase activity have recently emerged in the development and maintenance of the proteome, and impact telomere elongation, DNA repair, metabolism, and apoptosis. This paper traces the distribution of tankyrases throughout the branches of multicellular animals and identifies the single-celled choanoflagellates as a potential early origin of tankyrase proteins. The authors further characterized an anciently diverged conserved tankyrase homolog from the sponge, and emphasized that although catalytic function is conserved, interaction partners may be highly variable between species, as a potential etiology of their diverse repertoire of PARylated and non-PARylated protein targets.
The article is clear and well-written and the sequence homology analyses of ARC and SAM domains of the 113 tankyrase proteins, as well as the experimental functional validations are well done. The discussion is clear, brief, and interesting, These sequence analyses will be a useful contribution to our general understanding of proteostasis and molecular evolution. The article should be published without delay, and perhaps the discussion may be strengthened about if some hypotheses could be elaborated on how this conserved sequence information might inform us on how Tankyrases might exert control over multiple signaling pathways through both PARP-dependent and PARP-independent mechanisms.Something to the discussion perhaps
We thank the reviewer for the careful assessment of the manuscript. We have elaborated on some points in the discussion and extended the conclusion section.
The differences of PARP-dependent and PARP-independent functions of tankyrases are certainly an interesting aspect of their role as regulatory proteins, however the PARP-independent “scaffolding” functions currently lack sufficient experimental data and models. As such, we decided to not focus on this aspect in as it would mostly be speculation.
Reviewer 2 Report
1:The article is structured nicely and well written. I insist the authors to provide software used to identify potential interacting proteins with Tankyrase and to provide a reference for the same .
2:Elaborating in the Introduction regarding interaction of Tankyrase with their partner proteins wnt catenin pathway proteins and telomerases can by suggested.
Author Response
Reviewer #2
1:The article is structured nicely and well written. I insist the authors to provide software used to identify potential interacting proteins with Tankyrase and to provide a reference for the same .
We thank the reviewer for the careful assessment of the manuscript.
The list of human tankyrase binding proteins was manually curated based on experimental evidence for the specific TNKS-TBM interactions found in the literature. The references for this are shown in Table 2. We have now elaborated this in our methods section (section 2.5.).
2:Elaborating in the Introduction regarding interaction of Tankyrase with their partner proteins wnt catenin pathway proteins and telomerases can by suggested.
We chose to not elaborate on specific binding partners in the introduction to keep it focused, however the functions of TNKSs in Wnt signaling and in telomere homeostasis are also discussed in the discussion section (section 4.3.).